# Nanoemulsions of Hydroxysafflor Yellow A for Enhancing Physicochemical and In Vivo Performance

**DOI:** 10.3390/ijms24108658

**Published:** 2023-05-12

**Authors:** Yingjie Zhang, Cailing Zhong, Qiong Wang, Jingqing Zhang, Hua Zhao, Yuru Huang, Dezhang Zhao, Junqing Yang

**Affiliations:** 1Chongqing Key Laboratory of Biochemistry and Molecular Pharmacology, Chongqing Medical University, Chongqing 400016, China; 2Chongqing Research Center for Pharmaceutical Engineering, Chongqing Medical University, Chongqing 400016, China

**Keywords:** hydroxysafflor yellow A, oral administration, nanoparticle, cerebral ischemia-reperfusion injury

## Abstract

Stroke was always a disease that threatened human life and health worldwide. We reported the synthesis of a new type of hyaluronic acid-modified multi-walled carbon nanotube. Then, we produced hydroxysafflor yellow A-hydroxypropyl-β-cyclodextrin phospholipid complex water-in-oil nanoemulsion with hyaluronic acid-modified multi-walled carbon nanotubes and chitosan (HC@HMC) for oral treatment of an ischemic stroke. We measured the intestinal absorption and pharmacokinetics of HC@HMC in rats. We found that the intestinal absorption and the pharmacokinetic behavior of HC@HMC was superior to that of HYA. We measured intracerebral concentrations after oral administration of HC@HMC and found that more HYA crossed the blood–brain barrier (BBB) in mice. Finally, we evaluated the efficacy of HC@HMC in middle cerebral artery occlusion/reperfusion (MCAO/R)-injured mice. In MCAO/R mice, oral administration of HC@HMC demonstrated significant protection against cerebral ischemia-reperfusion injury (CIRI). Furthermore, we found HC@HMC may exert a protective effect on cerebral ischemia-reperfusion injury through the COX2/PGD2/DPs pathway. These results suggest that oral administration of HC@HMC may be a potential therapeutic strategy for the treatment of stroke.

## 1. Introduction

Stroke remains a common brain disease that generates devastating neurological deficits. It has a high mortality and causes major disability and remains a major public health challenge worldwide [1]. Stroke is divided into hemorrhagic stroke (with the rupture of cerebral blood vessels) and ischemic stroke (blockage of cerebral blood vessels); the latter accounting for about 87% of the total incidence of stroke [2]. Ischemic stroke is a kind of cerebrovascular accident caused by the reduction in or interruption of local blood supply to brain tissue [3], causing a series of complex biochemical molecular cascade reactions [4], such as energy metabolism disorders, activation of excitotoxic glutamate signaling pathway, free radical response, inflammatory response and so on, leading to neurological deficit. Treatment is aimed at restoring perfusion and reoxygenation. However, perfusion and reoxygenation can result in cerebral ischemia-reperfusion injury (CIRI) [5].With characteristic high incidence, high disability rate and high mortality, mechanisms such as excitotoxicity, oxidative stress-induced damage, apoptosis and inflammation may be involved in the process of ischemia-reperfusion [6,7]. Studies showed that inflammatory/immune responses play a key role in the pathogenesis of CIRI, and plays a dual role in promoting tissue damage and also repair. Therefore, inhibiting neuroinflammation induced by cerebral ischemic stroke became an attractive strategy in stroke therapy [8]. 

In recent years, a number of active ingredients in traditional Chinese medicine were described by researchers because of their effectiveness, safety and economy in the treatment of ischemic stroke. Safflower is a kind of traditional Chinese medicine, with the effect of promoting blood circulation and removing blood stasis. Safflower contains a variety of chemical components. Hydroxysafflor yellow A (HYA) is the main constituent of the yellow pigment by weight and has the greatest pharmacological effect. Many authors highlighted pharmacological activities of HYA including antioxidant [9], anti-tumor [10] and anti-ischemia reperfusion injury implicit in the effects [11]. Studies showed that HYA can protect against CIRI by a variety of mechanisms; for example, by preventing inflammatory reaction [12], relatively restoring the metabolic balance of glycolysis [13], inhibiting the development of oxidative stress [14] or by antagonizing the platelet activation [15]. However, HYA is a hydrophilic compound with a chalcone glycoside structure, which has poor lipid solubility and low oral bioavailability, a biopharmaceutics classification system (BCS) class III drug [16]. The blood–brain barrier (BBB) is the main barrier for brain disease interventions [17], but is poorly permeable to HYA. In regard to the current safflower yellow injection, which contains approximately 80% HYA, it received approval by the State Food and Drug Administration of China for the treatment of cardiovascular diseases such as angina pectoris [18]. However, an oral preparation of HYA for brain disease intervention is urgently required.

Nanoemulsions are biphasic dispersions of two immiscible liquids: either water in oil (W/O) or oil in water (O/W); stabilized by an amphiphilic surfactant [19]. Nanoemulsions consist of nano-scale droplets, which can slowly release, leading to accurate targeting with low toxicity. They increase drug contact area with the gastric mucosa, enhancing absorption, permeability of the BBB and prevent hydrolysis of the wrapped substance or the degradation of enzyme [20,21]. Nanoemulsions are promising drug carriers in brain-targeted preparations [22,23]. Desai et al. prepared oral darunavir-loaded lipid nanoemulsions, improving its bioavailability and enhancing brain to absorption [20]. Phospholipids are amphiphilic, with good fat solubility and biofilm compatibility. Drugs can be combined with phospholipids to prepare phospholipid complexes, thereby improving their bioavailability [24,25]. Results by Liu et al. showed that the stability constant of the complex formed with HPCD was enhanced after flavanones were complexed with cyclodextrin [26]. Chitosan is a cationic polysaccharide which adheres to mucous membranes, which can reversibly open Caco-2 cells to promote drug absorption [27]. Ma et al. prepared a HYA-CS complex, which increased the bioavailability of HYA 4.76 fold [28]. Lv et al. also found that HYA is a P-glycoprotein substrate analog, which is readily excreted by P-glycoprotein in the intestinal tract, leading to a strong bile efflux effect and low absorption [29]. 

Because the effect of compound formulations is often superior to that of single formulations, we considered the advantages of delivery of HYA with multiple carriers. Therefore, we produced HYA complexes with phospholipid and HPCD under various conditions, and then, we prepared water-in-oil HYA complex nanoemulsions with hyaluronic acid-modified multi-walled carbon nanotubes and chitosan (HC@HMC) by titration, aiming to improve its oral bioavailability and ability to cross the BBB for treatment of ischemic stroke via oral administration. We measured intestinal absorption and pharmacokinetics of HC@HMC in rats. We determined the intracerebral concentration of HC@HMC after oral administration. Finally, we explored the protective effect of HC@HMC on cerebral ischemia-reperfusion in mice and examined if HC@HMC may protect mice against cerebral ischemia-reperfusion injury through the COX2/PGD2/DPs pathway.

## 2. Results

### 2.1. Characterization of HC@HMC

The average Zeta potential of HC@HMC was approximately 0 mV and the particle size graph showed no results. It was seen from its appearance (Figure 1) that HC@HMC was a clear, uniformly dispersed liquid, demonstrating its successful preparation.

### 2.2. HC@HMC Improved the Oral Absorption of HYA Shown by Intestinal Perfusion in Rats 

Compared with HYA, HC@HMC showed higher absorption in total-intestinal tracts (duodenum, jejunum, ileum and colon). HYA and HC@HMC were mainly absorbed in the duodenum and colon. The measured Ka, Peff and absorption percentage (PA) values comparing HYA with HC@HMC the same gastro-intestinal areas were further described as follows.

The duodenal Ka value of HC@HMC was highest in each of the intestinal segments, while the colonic Ka value of HYA was the highest. The duodenal and ileal Ka values of HC@HMC were 17.82 × 10^−5^ s^−1^ and 7.18 × 10^−5^ s^−1^, which were, respectively, 15.10 times and 10.99 times that of free HYA. The colonic Peff values of HC@HMC and HYA were the highest of the intestine, respectively, 57.28 × 10^−5^ cm·s^−1^ and 1.93 × 10^−5^ cm·s^−1^. The colonic Peff value of HC@HMC were about 29.68 times of free HYA. The duodenal PA value of HC@HMC was the highest of the intestinal segments, while the colonic PA value of HYA was the largest. The PA of HC@HMC in the duodenum and colon was 79.41% and 79.02%, respectively, 48.71 and 11.19 times that of HYA (Figure 1A).

### 2.3. Pharmacokinetics of HC@HMC

The main pharmacokinetic parameters of the atrioventricular model are shown in Figure 1. It can be seen from Figure 1B that within 0–4 h, the blood concentration of HC@HMC was significantly higher than that of HYA. Due to its low solubility and poor permeability, the maximum blood concentration (Cmax) of HYA was only 1.25 ± 0.33 mg·L^−1^, while the Cmax of HC@HMC was as high as 20.14 ± 4.23 mg·L^−1^, which was 16.11 times that of HYA. In addition, the AUC_0–72h_, Tmax and MRT of HC@HMC were higher than those of HYA, and all parameters except Tmax and MRT_0–96h_ were significantly different. The AUC_0–96h_ and Cmax of HC@HMC were statistically about 10 and 16 times those of HYA, respectively, indicating that HC@HMC significantly increased the absorption of HYA in rats. In addition, the Cl of HYA was more than double that of HC@HMC, indicating that HC@HMC significantly increased the residence time in rats, showing enhanced absorption. The pharmacokinetic parameters of HC@HMC calculated by the non-compartment model were superior to HYA (Figure 1C), demonstrating superior pharmacokinetic behavior. 

### 2.4. Intracranial Concentration of HYA in Mice

To investigate whether our preparation could penetrate the BBB, we used (LC/MS) to determine intracranial drug concentrations in mice after administration. The results showed that both HC@HMC and HYA can penetrate the BBB; HC@HMC could penetrate the BBB to a greater extent. After 1 h of oral administration, the HC@HMC reached a peak concentration of 23.46 ng·mL^−1^, which was 1.15 times that of HYA (Figure 2A). We speculated that the water-in-oil nanoemulsion improved fat solubility, as it penetrated the BBB. Combining the blood drug concentration curve and the intracranial drug concentration curve, the trend of change and peak time of the HC@HMC was almost the same (Figure 1B). To further understand the blood–brain barrier permeability of the drug, the brain/plasma ratio of the drug was further calculated based on the above pharmacokinetic parameters (Figure 2B). All of the above results suggest that HC@HMC improves intestinal absorption and then penetrates the BBB more easily.

### 2.5. Effect of HC@HMC on Cerebral Injury Caused by MCAO/R in Mice

To determine whether HC@HMC affected CIRI, the TTC assay was used to determine brain infarct size. Representative images of TTC-stained brain sections are shown in Figure 3A. It is clear that, compared with the sham group, infarct volume was significantly increased in the MCAO/R group. Compared with MCAO/R group, the 80 mg·kg^−1^ and 160 mg·kg^−1^ HC@HMC groups and the 20 mg·kg^−1^ HYA group significantly decreased the infarct volume and neurological score (Figure 3B,C). Compared with the sham group, the residence time of mice on the rotating rod was significantly reduced in the MCAO/R group, while the 80 mg·kg^−1^ HC HMC and 160 mg·kg^−1^ HC@HMC groups and the 20 mg·kg^−1^ HYA group demonstrated significantly improved motor function and coordination of mice after surgery (Figure 3D).

To determine whether the brain showed histopathological changes, HE staining was conducted. In the sham group, nerve cells were arranged tightly in order, and most cells were round or oval with large cell bodies. However, in the MCAO/R and vehicle groups, the 40 mg·kg^−1^, nerve cells presented significant nuclear pyknosis, vacuolization and disordered arrangement (Figure 4A,C). Compared with the MCAO/R group, the 80 mg·kg^−1^ and 160 mg·kg^−1^ HC@HMC groups and the 20 mg·kg^−1^ HYA group demonstrated reduced nuclear deep staining vacuolization and mortality of nerve cells (Figure 4B,D).

### 2.6. HC@HMC Reduced Inflammation in MCAO/R Treated-Mice

We examined whether HC@HMC affected COX2, DP1 and DP2 expression. Compared with the sham group, expression of COX2, DP1, DP2 mRNA and protein were significantly increased in the cortex of MCAO/R-treated mice. Compared with the model group, HC@HMC markedly decreased the expression of COX2, DP1, DP2 mRNA and protein in the cortex of MCAO/R-treated mice, and HYA had the same effect (Figure 5A–F). To determine whether related products and inflammatory factors had changed, ELISA assay was used to determine PGD2, TNF-α and IL-1β content in mouse cortex. Compared with the sham group, the mice cortical PGD2, TNF-α, IL-1β concentrations were significantly increased in the model group. Compared with the model group, HC@HMC significantly blunted the increase in mouse cortical PGD2, TNF-α and IL-1β content, and HYA had the same effect (Figure 5H–J). 

## 3. Discussion

In general, drugs for the treatment of stroke must cross the BBB to prove effective. Studies showed that the gastrointestinal absorption and oral bioavailability of HYA are low [16]. Therefore, HYA is usually administered by intravenous administration. It follows that there is an urgent need for a solution to the poor blood stability and low BBB penetration of HYA. 

In this study, we produced HYA complexes with phospholipid and HPCD under specific conditions, and then, we prepared the water-in-oil HYA complex nanoemulsion with hyaluronic acid-modified multi-walled carbon nanotubes and chitosan (HC@HMC) by titration. We showed that HC@HMC significantly improved the absorption, increasing its bioavailability; it likely reduces and improves the efflux by P-glycoprotein and improve its absorption. Because polyethylene glycol 400 acts as a P-glycoprotein (P-gp) inhibitor, it promotes the absorption of the P-gp substrate HYA. We first prepared HYA with phospholipids and HPCD as a complex. The phospholipids were aimed to increase cell membrane fluidity, to assist opening of the tight junctions between cells [30,31], which enhances the permeability of hydrophilic drugs. Chitosan is also able to enhance drug permeability across BBB by affecting the tight junction [32]. HPCD can reduce the degradation of, and improve the stability of, HYA [26], enhancing and improving its bioavailability and activity. HC@HMC significantly increased the Cmax of HYA and prolonged MRT, thereby significantly increasing its AUC and improving its bioavailability. 

In addition to increasing drug plasma concentrations, effective BBB penetration is essential for effective stroke therapy. One hour after oral administration of HC@HMC, the peak concentration of HC@HMC was 23.46 ng/mL, which was 1.15 that times of HYA. Therefore, oral administration of HC@ HMC can achieve the same blood HYA concentrations as obtained by intravenous administration. The oral HYA preparations that we produced are likely to improve compliance of patients in clinical practice. The water-in-oil nanoemulsion is an effective strategy for oral delivery of highly water-soluble and low-permeability drugs, and for traversing the BBB.

Finally, we investigated the therapeutic effect of HC@HMC in a MCAO/R mouse model of ischemic stroke. Li sun et al. found that injecting HYA (1, 5, 10 mg·kg^−1^) into the tail vein 30 min before surgery protected rats from focal cerebral CIRI through inhibition of I/R-induced protein oxidation and nitration [14]. In another study, using the MCAO model, Lu Yu et al. found that the administration of 8 mg·kg^−1^ and 16 mg·kg^−1^ HYA by CCA injection improved impaired cognitive function in Morris water maze (MWM) testing as well as passive avoidance tasks in rats [33]. In clinical practice, the recommended dose of HYA is 100 mg (approximately 2 mg·kg^−1^) once daily for common cases or twice a day for severe cardiac patients [34]. In order to mimic the current clinical treatment regimen, we chose to treat the ischemic mice with HYA at a dose of 20 mg·kg^−1^ since, in our experiments, the intravenous injection of HYA 6 mg·kg^−1^ gave an AUC_0–24h_ of 23.85 ± 0.21 mg·h·L^−1^ and oral HC@HMC 60 mg·kg^−1^ gave an AUC_0–24h_ of 54.48 ± 9.87 mg·h·L^−1^. The calculated absolute bioavailability of HC@HMC was 23%, giving an effective dose of HC@HMC of 87 mg·kg^−1^. For ease of calculation, we chose a dose of 80 mg·kg^−1^. We also used a low dose of 40 mg·kg^−1^ and a high dose of 160 mg·kg^−1^. Our experimental results demonstrate that 80 mg·kg^−1^ and 160 mg·kg^−1^ HC@HMC and 20 mg·kg^−1^ HYA were all effective and suggest a certain dose-dependence. These results indicate that HC@HMC increased the bioavailability of HYA, thereby protecting mice from focal cerebral CIRI.

Prior to the investigation of mechanisms, three doses (40, 80 and 120 mg·kg^−1^) of HC@HMC were assessed according by neurological function scoring. The low-dose group showed no effect but there were clear improvements in neurological scoring in the two high-dose groups. However, there were no statistically significant inter-group differences. Therefore, we selected the 80 mg·kg^−1^ dose group to study the potential mechanism of action of HC@HMC. Compared with the sham group, the expression of COX2, DP1 and DP2 in the model group were significantly increased. Inflammation plays an important role in the occurrence and development of central nervous system diseases [35]. COX2 is a key rate-limiting enzyme in the inflammatory pathway [36]. PGD2 was downstream product of COX2, and is abundant in the brain [37]. Studies reported that some harmful effects of PGD2 are mainly regulated by DP2 receptors or through the PPARγ pathway [38]. Compared to the model group. HC@HMC 80 mg·kg^−1^ significantly reduced the expression of COX2, DP1 and DP2. This suggests that HC@HMC may protect mice against cerebral ischemia-reperfusion injury through the COX2/PGD2/DPs pathway. HYA has the same effect, suggesting that HC@HMC and HYA have the same mechanism. However, it was not clear whether HYA exerts its protective effect through the drug substance or its metabolites, and further research is needed. HYA was reported to have eight main metabolites in rats, but we do not know which plays an important role [39]. Furthermore, high-quality studies are required to examine this and to provide more possibilities for the treatment and improving the prognosis of ischemic stroke.

In summary, we developed an effective strategy to enable the penetration of HYA through the BBB with high stability, ameliorating cerebral ischemia-reperfusion injury in mice. These results suggest that our HC@HMC nanomedicine has important clinical therapeutic potential in stroke, owing to the ease of formulation, stability and BBB permeability.

## 4. Materials and Methods

### 4.1. Materials 

Hydroxysafflor yellow A (HYA) (98% purity) was purchased from Zhejiang Yongning Pharmaceutical Co., Ltd. (Taizhou, China), rutin was purchased from Chengdu Munster Biotechnology Co., Ltd. (Chengdu, China), hydroxypropyl-β-cyclodextrin from Jiangsu Taixing Xinxin Pharmaceutical Accessories Co., Ltd. (Taixing, China), phospholipids from Shanghai Liberty Biotechnology Co., Ltd. (Shanghai, China), chitosan from Zhejiang Jinhuo Biochemical Co., Ltd. (Wenzhou, China), glycerol monocaprylate (GMC) from Henan Zhengtong Food Technology Co., Ltd. (Xingyang, China), polyethylene glycol 400 from Sinopharm Group Chemical Reagent Co., Ltd. (Shanghai, China) and polyoxymethylene-hydrogenated castor oil (RH40) from Nanjing Durai Biotechnology Co., Ltd. (Nanjing, China). Poloxamer 188 and 407 were obtained from Nanjing Well Biochemical Co., Ltd. (Nanjing, China). Hyaluronic acid-modified multi-walled carbon nanotubes were produced in our own laboratory. Methanol and acetonitrile (chromatographically pure) were purchased from America TEDIA Co., Ltd. (Fairfield, OH, USA), and the remaining reagents were analytically pure. 

### 4.2. HPLC Analysis of HYA In Vivo

Chromatographic analysis was performed by HPLC, using the LC-20A HPLC system and UV-VIS 3150 ultraviolet-visible spectrophotometer (Shimadzu Corporation of Japan, Kyoto, Japan), with a Phenomenex C18 (250 mm × 4.6 mm, 5 µm) column. Detection wavelength was 403 nm. Acetonitrile (A)-20 mmol·L^−1^ potassium dihydrogen phosphate solution (B), pH = 3.5, were used for gradient elution for plasma and intestinal perfusion samples. The column oven temperature was 30 °C and flow rate was 1 mL·min^−1^. Rutin was used as an internal standard according to the manufacturer’s instructions. 

### 4.3. Synthesis of MWCNT-HA

Firstly, for synthesis of carboxylated multi-walled carbon nanotubes (MWCNT-COOH), a mixed acid solution (3:1 volume ratio of concentrated sulfuric acid and concentrated nitric acid) and 30% H_2_O_2_ solution (10 mL) were added into the MWCNT (100 mg). After 3 h ultrasonic reaction, it was diluted with a large amount of distilled water, using a 0.22 μm microporous filter membrane (mixed cellulose grease) for sucking and filtering, and then, it was repeatedly washed with distilled water to make the PH neutral and the filtered cake was placed in an oven at a constant temperature of 80 °C for drying, to yield MWCNT-COOH.

Secondly, for synthesis of aminated HA-NH2(HA), 200 mg of HA was dissolved in 10 mL formamide at 50 °C, EDC (520 mg) and NHS (310 mg) were dissolved in 10 mL and 5 mL of formamide and the activation reaction was performed for 30 min. An amount of 2 mL of ethylenediamine was then dissolved in 10 mL of formamide. Under ice bath conditions, the activated HA solution was slowly dripped into the ethylenediamine solution, controlling the rate (after 60 min of dripping). The reaction solution was raised to room temperature and the reaction continued for 3 h. Then, a large volume (4 times that of the original solution) was added to pre-chilled acetone precipitate, the resulting precipitate was filtered reconstituted with water, using a 0.45 µm filter membrane, and transferred to a dialysis bag (MWCO = 3500) for 48 h, and the water was changed every 6 h, before eventually being freeze-dried to obtain HA-NH2.

Finally, MWCNT-COOH was combined with HA using the amidation reaction of HA-NH2 and MWCNT-COOH under EDC and NHS conditions. Briefly, 30 mL of formamide was added to a beaker containing MWCNT-COOH (80 mg), and after 30 min of sonication under ice bath conditions, rinsing with formamide (20 mL), and the formamide was transferred to a round bottomed flask. EDC (305 mg) and NHS (182 mg) were dissolved in 5 mL formamide, and then, they were transferred to the round-bottomed flask, stirred and activated at room temperature for 30 min. A total of 160 mg HA-NH2 was dissolved in 10 mL formamide, and 180 μL of triethylamine was added; once activation of MWCNT-COOH was complete, rapidly added dropwise, and the reaction was carried out at room temperature for 24 h. Pre-cooled excess acetone (4 times the reaction solution) was added in an ice bath, cooled, crystallized and left to precipitate MWCNT-HA. This was then sucked and filtered using a 0.22 μm organic membrane, the precipitate was washed with acetone, reconstituted with ultrapure water, dialyzed in water using a dialysis bag (MWCO = 12,000 kDa) and, after 48 h, freeze-dried to obtain the final product MWCNT-HA. 

### 4.4. Preparation of HC@HMC

HYA, HPCD and PC (molar ratio 1:2.33:2.47) were dissolved in a clean and dry round-bottomed flask in an appropriate amount of absolute ethanol and stirred for 3.5 h (at 50 °C) in the dark. Rotary evaporation removed the absolute ethanol to obtain the HYA compound hydroxysafflor yellow A complex (HYAC); HYAC, GMC, RH40 and PEG400 (mass ratios 1:0.31:6.77:6.77) were magnetically stirred until the HYAC was completely dissolved, and then, the Chitosan solution (2 mg·mL^−1^) was added dropwise. The solution underwent a change from turbid to clear and transparent. Finally, a certain amount of novel hyaluronic acid-modified multi-walled carbon nanotubes (HMC),with F188 and F407 (mass ratio of 1:5:5) mixture was added, and stirring was continued until dissolution was complete, forming a clear, uniformly dispersed thick viscous liquid, consisting of hydroxysafflor yellow A-hydroxypropyl-β-cyclodextrin phospholipid complex water-in-oil nanoemulsion with hyaluronic acid-modified multi-walled carbon nanotubes and chitosan modified (HC@HMC). 

### 4.5. Characterization of HC@HMC

A Malvern laser particle size potentiometer (Zetasizer Nano zs90; London, UK) was used to determine the particle size and Zeta potential of HC@HMC; a conductivity meter (DDS-307A; Shanghai, China) was used to determine the conductivity and pH of the HC@HMC.

### 4.6. Absorption in the Intestine

The in situ absorption of HYA and HC@HMC by the intestinal system of rats was investigated using the unidirectional perfusion method [40]. All SD rats were raised under controlled conditions and fasted over more than 12 h before the drug was administered. Rats were anesthetized by intraperitoneal injection of 20% uratan solution (7 mL·kg^−1^). Then, the rats were fixed and a midline abdominal incision was made. The duodenum, jejunum, ileum and colon were located, and a small cannula was inserted at the upper end of the four intestinal segments, ligated and fixed at the lower end, and the intestinal segment corresponding to each cannula was marked and the wound was covered with clean gauze soaked with saline to keep it moist. The contents were rinsed with physiological saline at a constant temperature of 37 °C, drained, and then equilibrated with Krebs-Ringer solution at a flow rate of 0.25 mL·min^−1^ for 15 min. Finally, HYA or HC@HMC perfusion fluid (placed in a 100 r·min^−1^ magnetic stirrer) was injected and stirred at a flow rate of 0.25 mL·min^−1^. After 1 h, the perfusate outflow at each of outlets of the four intestinal segments was collected individually. Following the perfusion experiment, the four intestinal segments were excised and the length (L) and radius (R) of each segment was measured and recorded. The concentration of HYA in the intestinal perfusates was determined by HPLC. Intestinal fluid samples were sonicated with methanol for 20 min, and the continuous filtrate filtered by a 0.45 μm microporous membrane was collected. For the 100 μL intestinal juice sample, 10 μL 50 μg·mL^−1^ RT and 50 μL perchloric acid solution (1 mol·L^−1^) were added and vortexed at 3000 rpm for 3 min, and centrifuged at 12,000 rpm for 10 min, before being analyzed by HPLC.

The measured HYA content in the intestinal perfusion sample was input into the following three formulas to calculate the absorption rate constant (Ka), effective permeability coefficient (Peff) and absorption percentage (PA) of HYA and HC@HMC, respectively.
$$K_{a}=\frac{X_{0}-X_{t}}{C_{0}t\Pi R^{2}L}\left(\times 10^{-5},\;1\cdot s^{-1}\right)$$
$$P_{eff}\frac{Q\times L_{n}(\frac{X_{in}}{Xout})}{2\Pi PL}\left(\times 10^{-5},\;\mathrm{cm}\cdot s^{-1}\right)$$
$$PA=\frac{X_{0}-X_{t}}{X_{0}}\left(\%\right)$$

In this formula, X_0_ represents the total mass of the initial drug; X_t_ represents the total mass of the drug at time t; C_0_ represents the initial drug concentration; X_in_ represents the total mass of the drug entering the perfusate; X_out_ represents the total mass of the drug flowing out of the perfusate; t represents perfusion time; Q stands for flow velocity; R and L are the inner diameter and length of each perfused intestinal segment. Non-parametric Wilcoxon signed rank sum statistical testing test was used for the Ka, Peff and PA to compare the significant difference between HYA and HC@HMC (*p* < 0.05).

### 4.7. Pharmacokinetics of HC@HMC 

Twelve male SD rats weighing (230 ± 20) g were randomly divided into 2 groups and fasted for 12 h. Two groups were gavaged with HYA solution and HC@HMC (equivalent to HYA 6 mg·kg^−1^). After administration, 0.5 mL blood was collected from the orbit of rats at different time points. Each sample was immediately transferred to heparin-containing centrifuge tubes, and after centrifugation at 6000 r·min^−1^ for 10 min, the supernatant was taken for measurement. 10 μL 50 μg·mL^−1^ RT and 50 μL 6% perchloric acid solution was added to 100 μL plasma, vortexed for 3 min, centrifuged at 12,000 r·min^−1^ for 10 min, and 20 μL of supernatant for injection measurement was removed. Pharmacokinetic parameters were analyzed by DAS 2.1.1 using non-compartmental analysis, and then, the analysis of variance and double one-sided *t*-testing were carried out. DAS 2.1.1 is a pharmacokinetic data analysis software that can calculate pharmacokinetic parameters and bioequivalence from drug concentration-time data.

### 4.8. LC/MS Analysis of HYA in Brain

Eighteen mice were randomly divided into six groups. After intravenous administration of HYA solution (40 mg·kg^−1^) or oral administration of HC@HMC (160 mg·kg^−1^), the mice were sacrificed 0.5 h, 1 h or 1.5 h later, and brain tissue was taken and weighed. A total of 500 μL of ultrapure water was added into the homogenizer and brain tissue was homogenized; the homogenate was centrifuged at 4 °C (12,000 r·min^−1^, 10 min); then, 100 μL of supernatant was removed, 10 μL of internal standard solution and 490 μL of methanol (precipitate protein), recentrifuged, centrifuged in the same way after mixing, and the supernatant was used for sample injection onto the column.

Ultra High Performance Liquid Chromatography Tandem Mass Spectrometry (Agilent 6470, Agilent Technologies, Santa Clara, CA, USA) was used to detect intracranial HYA concentrations in mice after its administration, ACE Excel 2 C18-PFP columns (2.1 × 100 mm, 2.0 μm) were employed for separation, 0.1% formic acid-methanol was used as the mobile phase for gradient elution, the flow rate was 0.2 mL·min^−1^ and the injection volume was 5 μL. ESI+Agilent Jet Stream ion source was used with multi-reaction monitoring mode (MRM) detection. Rutin was used as the internal standard for quantification.

### 4.9. Effect of HC@HMC on Cerebral Injury Caused by MCAO/R in Mice

#### 4.9.1. Animals

C57BL/6 male mice aged 8–10 weeks (18 to 22 g) were purchased from the Laboratory Animal Center, Chongqing Medical University, China (license number: SYXK YU 2012-0001). The mice were housed under a temperature-controlled environment (22–26 °C) and humidity (40–70%) with a 12 h light-dark cycle. The mice were supplied with standard rodent chow and water. All experiments in this study were consistent with the National Institute of Health Guide for the Care and Use of Laboratory Animals.

#### 4.9.2. Middle Cerebral Artery Occlusion/Reperfusion Model 

Ischemic stroke was produced using the MCAO/R method as previously reported [41]. The mice were anesthetized by intraperitoneal sodium pentobarbital (40 mg·kg^−1^). The right common carotid artery (CCA), external carotid artery (ECA) and the internal carotid artery (ICA) were exposed under a surgical microscope. The whole of the ECA and CCA were ligated. A nylon suture (Jialing, Guangzhou, China) was inserted from the CCA into the right side of ICA in a depth of 10 ± 0.5 mm, to occlude the origin of the middle cerebral artery (MCA). After 1 h occlusion, reperfusion was achieved by withdrawing the suture to restore blood supply to the MCA territory. Sham-operated mice underwent the same surgery, but without suture insertion. Body temperature was maintained at 37 ± 0.5 °C during surgery by a heating blanket.

#### 4.9.3. Protocol

Mice were randomly divided into seven groups: (1) sham group (n = 10), (2) MCAO/R group (n = 10), (3) MCAO/R + vehicle group (n = 10), (4) MCAO/R + 40 mg·kg^−1^ HC@HMC group (n = 10), (5) MCAO/R + 80 mg·kg^−1^ HC@HMC group (n = 10), (6) MCAO/R + 160 mg·kg^−1^ HC@HMC group (n = 10), (7) MCAO/R + 20 mg·kg^−1^ HYA group (n = 10). Mice were given HYA (intravenous administration) and HC@HMC (intragastrically administration) once daily, 5 days before surgery. Mouse brains were removed at 24 h after MCAO/R operation. The clinically recommended dose of HYA is 100 mg (approximately 2 mg·kg^−1^) once a day for usual cases or twice a day for severe cardiac patients [34]. Therefore, to mimic the current clinical treatment regimen, we chose to treat the ischemic mice with HYA at a dose of 20 mg·kg^−1^. which related to absolute bioavailability and corresponded to a dose of 80 mg·kg^−1^ HC@HMC.

#### 4.9.4. Neurological Scoring

Evaluation of neurological deficit evaluation was conducted after 24 h-reperfusion according to the Zea Longa’s method as previously described (Longa et al. 1989) [42]. The criteria used were as follows: 0, normal, no neurological deficit; 1, mild neurological deficit, failure to completely extend the right forelimb; 2, moderate neurological deficit, twisting to the contralateral side; 3, severe neurological deficit, falling to the left; 4, no spontaneous motor activity. Animals that died during the process were not included in the statistics.

#### 4.9.5. Rota Rod Test

Motor coordination was tested by using the Rota-rod tread-mill test [43]. The Rota-rod apparatus (UGO BASILE S.R.L, Gemonio, Italy) consisted of a rotating rod (75 mm diameter), on which the mice were allowed to stand. After twice daily training for 2 days (at a speed of 10 rpm to 40 rpm over 5 min), mice were tested 2 times for preoperative value on the third day. After 24 h reperfusion, mice were tested 2 times for postoperative value. The time for each mouse to remain on the rotating rod was recorded. The maximum time allowed was 5 min. The apparatus automatically recorded the time of falling.

#### 4.9.6. 2,3,5-Triphenyltetrazolium Chloride Staining

A total of 24 h after reperfusion, the mice in all groups were anesthetized and decapitated, and the brains were removed and frozen in the refrigerator within 20 min. Brains were cut into coronal sections of 2 mm thickness. The slices were stained with 2%(*w*/*v*) 2,3,5-triphenyltetrazolium chloride (TTC) (Sigma, Milwaukee, WI, USA) solution at 37 °C in the dark and were then fixed in 4% (*v*/*v*) phosphate-buffered paraformaldehyde solution. The infracted areas appeared as pale staining, and the posterior surface of each section was measured using Image J software (NIH, Bethesda, MD, USA).

#### 4.9.7. Hematoxylin/Eosin (HE) Staining

Hematoxylin and eosin (HE) staining was used to observe pathological histological damage in the cerebral cortex and hippocampus. After 24 h reperfusion, the mice were anesthetized with sodium pentobarbital and perfused with PBS, and were then perfused with 4% paraformaldehyde. Then, the brains were removed, dehydrated with graded ethanol, embedded in paraffin and cut into 5-μm-thick sections. Finally, the sections were stained with HE reagents and observed by light microscopy.

#### 4.9.8. Real-Time Polymerase Chain Reaction

Total RNA was isolated from mouse cerebral cortices cerebral cortex by Trizol reagent (Vazyme, Nanjing, China) according to the manufacturer’s protocol. Reverse transcription of mRNA was performed using HiScript Q Select RT SuperMix (Vazyme, Nanjing, China). To detect the amount of COX2 mRNA and DP1 mRNA and DP2 mRNA. The SYBR Green II (Biomake, Houston, TX, USA) incorporation method was applied with ATGB being an internal control for mRNA. The primer sequences are reported in Table 1.

#### 4.9.9. Western Blotting

Mouse cortical samples were lysed with RIPA containing 1% phenylmethanesulfonyl fluoride (PMSF) (100 mM). The buffered lysis samples were collected into an EP tube and centrifuged at 12,000 rpm at 4 °C for 15 min. The precipitate was discarded and the protein concentration was determined using the BCA Protein Assay Kit (Beijing, China). We performed Western blotting as described previously. In short, proteins were separated and transferred onto a polyvinylidene fluoride (PVDF) membrane (Millipore, MA, USA). The membrane was incubated with primary antibodies overnight at 4 °C. After washing, the membranes were incubated with horseradish peroxidase-conjugated secondary antibodies (Proteintech, Wuhan, China) for 1h at room temperature. Finally, the immunoblots were developed by ECL Western blotting detection reagent (Millipore, USA). The primary antibodies against COX2 (dilution 1:1000, Abcam, Cambridge, UK) DP1 (dilution 1:1000, Abcam, USA) and were purchased from Abcam. DP2 (dilution 1:500, Aiffinity, Changzhou, China) was purchased from Aiffinity. Tublin (dilution 1:5000, Proteintech, China) was purchased from Proteintech.

#### 4.9.10. Enzyme-Linked Immunosorbent Assay

The levels of PGD2, IL-1ꞵ and TNF-α were detected by following the protocol of a commercial ELISA kit (MeiMian, Changzhou, China). Briefly, the sample, the standard and an HRP-labeled antibody were added to microwells coated with the antibody, incubated at 37 °C and washed thoroughly. The color was developed with a TMB substrate, which was converted to blue by peroxidase catalysis and finally converted to yellow by acid action. The intensity of the color was positively correlated with the expression of the target protein in the sample. The optical density value was measured with a microplate reader at a wavelength of 450 nm, and the sample concentration was calculated.

### 4.10. Statistical Evaluation

All data were reported as mean ± standard deviation. Statistical significance was determined by *t*-test to compare two groups or one-way analysis of variance (ANOVA), followed by Tukey’s multiple comparison test for multiple comparisons. Statistical analysis was performed by GraphPad Prism 6 (GraphPad Software, Boston, MA, USA). A *p* value of less than 0.05 was regarded as statistically significant.

## Data Availability

The data presented in this study are available on request from the corresponding author.

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
