# Peer review of "Nanoemulsions of Hydroxysafflor Yellow A for Enhancing Physicochemical and In Vivo Performance"

_ijms, 2023, doi:10.3390/ijms24108658_

Round 1
Reviewer 1 Report (Previous Reviewer 1)
This manuscript investigated the therapeutic effect of novel nanoemulsions of hydroxysafflor yellow A (HYA) in middle cerebral artery occlusion/reperfusion (MCAO/R)-treated mice. They also determined the pharmacokinetic parameters of the formulation and conducted mechanistic studies.
There are some concerns:
1) The author showed HPLC analysis of HYA in vivo but did not show any validation studies for the method.
2) The author did not show any toxicity studies for prepared HC@HMC.
3) The authors did LC/MS analysis of HYA in brain tissue from mice but PK studies in rats. The authors did not justify this part. Which part of the brain was used for the drug study?
4) No explanation was provided for the vastly improved effectiveness. I cannot agree that this is attributable to increased delivery, particularly because the increased delivery was not convincingly shown. A 1.15-fold increase in delivery did not account for the drastic decrease in stroke volume.
Author Response
Please see the attachment.

Reviewer 2 Report (Previous Reviewer 2)
• A brief summary
In the present paper is presented : the prepared hydroxysafflor yellow A-hydroxypropyl-β-cyclodextrin phospholipid complex water-in-oil nanoemulsion with hyaluronic acid-modified multi-walled carbon nanotubes and chitosan (HC@HMC) for oral treatment of ischemic stroke.The main contribution of this paper is that : Was found that the intestinal absorption and the pharmacokinetic behavior of HC@HMC is better than HYA ( = hydroxysafflor yellow A)
• Generalities for the present research article:
• The manuscript is clear and presented in a well-structured manner
• The cited references are in between 1989 and 2020, are relevant even there are studies on this material HYA presented after 2020 in the literature. The manuscript Does not include self-citations.
• The manuscript scientifically sound interesting and the experimental design is very interesting to test the hypothesis
• Are the manuscript’s results reproducible based on the details given in the methods section? – yes.
• The the figures/tables/images/schemes are interesting.
• was explain at page 4-5 in the formula of Ka and Peff the term ( E-05, ls-1) and ( E-05, cm.s-1)
• AT page 5 please explaine what is DAS 2.1.1.
• At page 17 is 2 of 23 at row 16 and 17 :” Therefore, there is an urgent need for powerful nanomaterials to overcome the 16 problems of poor blood stability and low BBB penetration efficiency of HYA” , you can make a ADME simulations and the results is no BBB penetration , how can comment this?
• Are the conclusions consistent with the evidence and arguments presented?
• In paragraph 4 they have to write Discussion and Conclusions to have a response more consistent
• The conclusions are strong and consistent with the evidence and arguments presented.
• There are a lot of results (too much results) but not so many discussion
• By evaluate the ethics statements and data availability statements they are adequate: in the paragraph 2.9.1. Animal: is stated: “All experiments in this study were consistent with the National Institute of Health Guide for the Care and Use of Laboratory Animals “. More there is no conflict of interest between authors.
• The review is clear, comprehensive and of relevance to the field. Can be a gap in knowledge.
• The cited references are in between 1989 and 2020, are relevant even there are studies on this material HYA presented after 2020 in the literature. The manuscript does not include self-citations. This current review is still relevant and of interest to the scientific community
Rating the Manuscript
• Novelty: The present research is original and well-defined. The results provide an advancement of the current knowledge.
• Scope: The present work fit the journal scope.
• Significance: The interpretation of the results must be improved. They are significant. A lot of results but not so much interpretations.
• Quality: The quality of presentation must be improved, we need more comments and discussion related of other work.
• Scientific Soundness: This study is correctly designed and technically sound. The analyses were performed with the highest technical standards. The data are robust enough to draw conclusions. (but they need much more comments ) . That the methods, tools, software, and reagents are described with sufficient details to allow another researcher to reproduce the results.
• Interest to the Readers: The conclusions are strong and interesting for the readership of the journal. The paper will attract only to a limited number of people, those interested in this substances and that method.
• Overall Merit: Yes there si an overall benefit to publishing this work. And this work advance the current knowledge.
• English Level: The English language can be improved!!!!
Round 2
Reviewer 1 Report (Previous Reviewer 1)
The author made significant improvements to the manuscript, which is suitable for publication.
This manuscript is a resubmission of an earlier submission. The following is a list of the peer review reports and author responses from that submission.
Round 1
Reviewer 1 Report
This manuscript investigated the therapeutic effect of novel nanoemulsions of hydroxysafflor yellow A (HYA) in middle cerebral artery occlusion/reperfusion (MCAO/R)-treated mice. They also determined the pharmacokinetic parameters of the formulation and conducted mechanistic studies.
There are some concerns:
1. How was the dosage of HYA solution and HC@HMC (equivalent to HYA 6 mg·kg-1) determined for PK studies? Also, for the brain, how was the dosage of HYA solution (40 mg·kg-1) or oral administration of HC@HMC (160 mg·kg-1) determined?
2. The authors did not show any dose-dependent toxicology studies for the HC@HMC nanoemulsions in mice.
3. The author did not demonstrate any efficacy or release profile studies for the prepared HC@HMC nanoemulsions.
4. The author conducted only one motor test (the Rodarod Test) for behavioral studies. The author should conduct one more test to validate this behavioral testing, like the cylinder or grid-walking test.
5. The author mentioned in the discussion section that "Prior to the study of mechanism, 3 doses (40, 80, and 120 mg·kg-1) of HC@HMC were assessed according to neurological function score." In the manuscript, the highest dose for the experimental trial was 160mg.kg-1.
6. The manuscript needs revision for language and grammar.
Author Response
Thanks for your recommendation and guidance.
1.Answer: It has been stated in the article that the minimum effective dose for rats is 2mg/kg, hich translates to 20mg/kg for the tail vein of mice. So when we did the drug efficacy experiment, the HySA was 20. Because the pharmaceutical formulations we studied were oral formulations, the final 80 is taken after the result of the conversion of the mode of administration is taken. Then when testing the concentration of drugs in the brain, when the free drug was 20mg/kg. We could not detect the drug concentration, because it was lower than the machine detection line. So we increased the dose, the free drug concentration was 40mg/kg, and the preparation concentration was 160mg/kg.
2.First, because the dose of HYA for gastric gavage of SD rats in this article is within the safe range. And numerous dose-related studies on HYA can confirm this. Secondly, the new formulation,HMC ,also has relevant literature to show its safety. Therefore, no dose-dependent toxicological studies were performed alone.
3.The in vitro release characteristics of HC@HMC do not necessarily reflect its effects in vivo. Moreover, the effect of HC@HMC in vivo has also been demonstrated in experiments on SD rats.
4.In this study, we only evaluated the motor function of mice. In many studies of brain injury, only the Rota Rod Test has been used to assess motor function. And cerebral ischemia-reperfusion also has a great impact on motor function, so the protective effect of HC@HMC has been verified with Rota Rod Test.
5.This part was written incorrectly, the dose of the preparation was designed with a double gradient, it should be "40, 80, 160".

Reviewer 2 Report
- Please explain at page 6 in the formula of Ka and Peff the term ( E-05, ls-1) and ( E-05, cm.s-1)
- AT page 6 please explaine what is DAS 2.1.1.
- At page 17 is 2 of 23 at row 16 and 17 :” Therefore, there is an urgent need for powerful nanomaterials to overcome the 16 problems of poor blood stability and low BBB penetration efficiency of HYA” , you can make a ADME simulations and the results is no BBB penetration , how can comment this?
- In paragraph 4 they have to write Discussion and Conclusions to have a response more consistent

Author Response
Thanks for your recommendation and guidance.
1.Answer:”E-05” means “10-5”.”l.s-1” and “cm.s-1” are units of Ka and Peff.
2.Answer:DAS 2.1.1 is a pharmacokinetic data analysis software that can calculate pharmacokinetic parameters and bioequivalence from drug concentration-time data.
3.Answer:In the studies on this topic, TTC tissue sectioning, the Rodarod Test, and Histopathology of hippocampus and cortex in mice have been used to verify that HC@HMC can penetrate the blood-brain barrier. Therefore, there were no more ADME simulations to illustrate this problem.
4.Answer:The relevant discussions and conclusions were already written in paragraph 4
Reviewer 3 Report
Overall: Zhang and colleagues present a manuscript showing a nano formulation of hydroxysafflor yellow A (HYA) for oral delivery to the brain and therapy of ischaemic stroke. There is a lot of data presented. The manuscript is well written, hypothesis and aims are clearly described. The paper is a very “standard” presentation of a new nano formulation, showing pK properties, drug absorption, drug delivery, and some animal model tests showing infarct area and analysis of brain function. I have one major concern with the overall findings. Please see the specific comments below.
Introduction: Some minor grammar issues to be tidied. The content is fine and rationale for the study is clearly spelled out.
Methods: Section 2.4 is too vague. “A certain amount” and “novel hyaluronic acid-modified multi-walled carbon nanotubes (HMC), F188 and F407 mixture”
I think the authors need to specify exactly what these components are. Methods need to be described so that others could replicate the work.
Section 2.8 I note that there is no mention of systemic perfusion. So the BBB penetration data should be interpreted with this in mind. There is likely over-estimation of brain penetrance if blood vessels were not flushed to remove free drug.
Results:
Figure 2. This figure should be one of the most important in the manuscript, since the authors’ hypothesis is based on improved brain delivery. But I see several problems with these data at the moment.
First, I am not sure how to interpret the Y axis of ng/mL. If we are measuring brain penetration, this would be better expressed as ng per unit mass, depending on how much brain tissue was homogenised for drug extraction. So please calculate actual drug delivery based on mass.
The text in section 3.4 also mentions 1.15x higher brain concentration of HC@HMC. However, the graph shows that the difference is not significantly different at any time point. Thus, such a claim cannot be made.
Second, it is useful to show the blood:brain ratio since your HM@HMC has very different pK properties to HYA. Figure 1 (and the text) shows a Cmax of 20-fold higher for HM@HMC compared to HYA. This means a 20-fold higher blood concentration produces only a 1.15x (and not significant) increase in brain concentration? To me that would indicate that the brain penetrance of the HM@HMC is extremely low and the authors hypothesis is not really supported.
Third, looking at actual concentrations, the blood concentration is around 10 µg/ml after 60 minutes, but the brain concentration is around 20 ng/ml. This means only ~0.2% of the drug is crossing the BBB. Combined with the lack of systemic perfusion to clear drug from brain microvessels, this is not very convincing.
Figure 3C, Y axis should say “infarct” not “infract”
The MCAO model data looks fine. But now I have to wonder where the huge reductions in infarct volume are coming from, since there is no evidence that the HMC concentration is any different.
Figure 4A, the images could be improved by enlargement and better quality (though maybe this is due to compression from the PDF generation). It is very difficult to make out details. Labelling would also help. The text claims “nerve cells presented significant nuclear pyknosis, vacuolization and disordered arrangement” but this is not easily visible in the H&E images. Can some arrows/labels be added to point out these morphological changes to readers?
Author Response
Thanks for your recommendation and guidance.
Answer:
Methods:
Here "a certain number of novel hyaluronic acid-modified multi-walled carbon nanotubes, F188 and F407 mixtures”, hyaluronic acid-modified multi-walled carbon nanotubes mean MWCNT-HA, and MWCNT-HA F188 and F407 were mixed at a mass ratio of 1: 5: 5. The relevant content has been modified in the text.
results-figure 2:
In Figure 2, the chart shows that 1 hour after oral HC@HMC, the HC@HMC reached a maximum concentration of 23.46 ng·mL-1, which is 1.15 times that of HYA. Based on this, we speculate that water-in-oil nanoemulsion improves fat solubility, making it more permeable to the blood-brain barrier.Although there was no significant difference, combined with the blood drug concentration curve and the intracranial drug concentration curve in Figure 1B, the trend and peak time of the Hc@HMC were almost the same. This result suggests that HC@HMC first promotes intestinal absorption and then more easily crosses the blood-brain barrier.
results-figure 3:
Because HMC is only a carrier of HC@HMC composite nanoemulsions,.HMC has an effect only on the pharmacokinetic process of HYA. The reductions in infarct volume of cerebral infarction is due to an increase in the dose of HC@HMC.
results-figure 4:
The black lines in the figure have marked which of the nerve cells belong to the nuclear abscess that exhibits significant nuclear abscess, vacuolization and disorderly arrangement.
Reviewer 4 Report
The manuscript developed a new hyaluronic acid nanotube and then put them into nanoemulsion to get HC@HMC that was tested for PK, intestinal absorption and cerebral ischemia-reperfusion injury (CIRI) treatment. However, there are many issues on formulation, study and manuscript writing. The formulation is very complex, but authors did not get any data for particle characterization. They actually did not know if they got nanoemulsion or not. This manuscript cannot be published.
1. Nanoemulsion is used to deliver HYA to cross the blood-brain barrier (BBB) by oral administration. Nanoemulsion might help drugs to cross the BBB by injection, but it will not help oral drugs. After oral absorption, oral drugs are taken as single molecule to enter blood circulation. There are no nano emulsions any more in the blood. How can nanoemulsion facilitate brain uptake. Thus, the overall design, rational and strategy of this HYA formulation is problematic.
2. Authors developed HPLC method to measure HYA in vivo samples. However, there are no data for HPLC method validation.
3. What is HYA? hydroxysafflor yellow A or MWCNT-HA ? what is HA? In Section 2.4, HYA was mixed with HPCD and PC. Where is MWCNT-HA in this procedure?
4. Overall, the formulation HC@HMC is too complicated and the study did not have a well controlled and analyzed preparation.
5. What is the instrument used in Section 2.t fo measure absorption int eh interline?
...
...
There are many other issues that are not described here.
Author Response
Answers:
1.Most of the nanopreparations are still mostly nanopreparations in the initial stage after oral administration into the blood, and some of them are free in the blood, because the clearance rate of nanopreparations in the blood is slower than that of free HYA, indicating that the drugs in the nanopreparations are not easy to be cleared or metabolized, and the retention time in the nanopreparations is longer. If all are free HYA after oral administration into the blood, the clearance rate will not vary greatly.
2.In the study of the pharmacokinetics of HC@HMC, we determined three indicators: Ka, Peff, and PA. The parameters used to calculate the above indicators are our results from HPLC and LC/MS.
3.
The full names of relevant abbreviations in the question are shown in the following table. MWCNT-HA, F108 and F407 were mixed evenly and finally added into the nanoemulsion prepared by HYA and HPCD complex, forming a clear, uniformly dispersed viscosity thick liquid, that was HC@HMC. "hyaluronic acid-modified multi-walled carbon nanotubes" in the "Preparation of HC@HMC" section of the text has been replaced with its abbreviation MWCNT-HA.
|
Abbreviations |
Full names |
|
HYA |
Hydroxysafflor Yellow A |
|
HA |
Hyaluronic Acid |
|
MWCNT-HA |
Hyaluronic Acid-Modified multi-Walled Carbon Nanotubes |
4.
Nanoemulsions composition is relatively complex, mostly to make a more stable, or implement more features, in order to improve the effect of drug treatment. In addition, all nanoemulsions contain oil phase, water phase and emulsifier1. The function of each excipients in HC@HMC prescription is shown in the following table, which includes the corresponding components. The experimental results showed that HC@HMC could improve the bioavailability of HYA and enhance the therapeutic effect of HYA. Therefore, each component in HC@HMC is reasonable, and each excipient was added in accordance with a certain ratio and sequence, which could better control the repeatable preparation of the preparation and achieve better results.
In addition, some excipients play a good role in enhancing oral drug absorption, which has been introduced in the introduction, such as phospholipids and Chitosan, phospholipids are amphiphilic lipids, which havewith good fat solubility in fat and biofilm compatibility. Drugs can be combined with phospholipids to prepare phospholipid complexes, thereby improving their bioavailability2,3. Chitosan is a cationic polysaccharide with the function of adhering mucous membrane adhesion function, which can reversibly open the tight connection between Caco-2 cells to promote drug absorption4,5.
The prescription of HC@HMC
|
Drug/Accessories |
Function |
|
Hydroxysafflor Yellow A (HYA) |
Main drug |
|
Phospholipid and Hydroxypropyl-β-cyclodextrin |
Carrier for HYA to prepare complex |
|
Glycerol monocaprylate (GMC) |
Oil phase |
|
Polyoxyethylene castor oil (RH/40) |
Emulsifier |
|
Polyethylene glycol 400 (PEG400) |
Co-emulsifier |
|
Chitosan solution |
Water phase |
|
MWCNT-HA |
Carrier |
|
F188 |
Emulsifier |
|
F407 |
Emulsifier |
Ref.
- Sun L, Yang L, Xu YW, et al. Neuroprotection of hydroxysafflor yellow A in the transient focal ischemia: inhibition of protein oxidation/nitration, 12/15-lipoxygenase and blood-brain barrier disruption. Brain Res.
- Abd-Elsalam WH, El-Helaly SN, Ahmed MA, Al-Mahallawi AM. Preparation of novel phospholipid-based sonocomplexes for improved intestinal permeability of rosuvastatin: In vitro characterization, dynamic simulation, Caco-2 cell line permeation and in vivo assessment studies. Int J Pharm.
- Wu H, Long X, Yuan F, et al. Combined use of phospholipid complexes and self-emulsifying microemulsions for improving the oral absorption of a BCS class IV compound, baicalin. Acta Pharm Sin B.
- Wang Y, Bai X, Hu B, et al. Transport Mechanisms of Polymannuronic Acid and Polyguluronic Acid Across Caco-2 Cell Monolayers. Pharmaceutics.
- Ma GN, Yu FL, Wang S, Li ZP, Xie XY, Mei XG. A novel oral preparation of hydroxysafflor yellow A base on a chitosan complex: a strategy to enhance the oral bioavailability. AAPS PharmSciTech.
5.The instrument used in the Section 2 was high performance liquid chromatograph, which was used to determine the HSA content in rat plasma or enema solution after administration. These measurements were then used to calculate pharmacokinetic parameters or intestinal absorption of HYA or HC@HMC.
Round 2
Reviewer 1 Report
The authors explained my concerns; therefore, I feel the manuscript is ready for publication with some English editing.
Author Response
Thank you for your affirmation.Please see the attachment.

Reviewer 3 Report
The authors made many changes to the manuscript, but it is mostly just tweaking sentence grammar and structure. In terms of actual scientific content, nothing is really amended. The authors did not provide a proper point-by-point response to the reviewer questions, as such, many of the points I raised were totally ignored or brushed aside.
I asked for brain delivery to be shown as blood:brain ratio and percentage delivery. Neither was done.
I asked for an explanation of Y axis units on Figure 2. It wasn't done.
No explanation for hugely increased efficacy was given. I can not accept that this is due to higher delivery - especially when the higher delivery was not robustly shown. A 1.15x increase in delivery could not explain a dramatically smaller stroke volume.
Author Response
Thanks for your kind suggestions.
Why the unit of figure 2 is “ng/ml”?Because HyA is well water-soluble, the tissue is taken at that time and it directly corresponds to adding water to dissolve, such as 200mg to add 200ml, so this is a proportional conversion process, and the machine calculates ng/ml. Secondly: to measure the blood concentration in the brain, we have performed systemic perfusion and then removed the brain tissue.
When we measured the concentration in the brain, the dose of intravenous HYA was 40 (mg/kg-1), the dose of oral HC@HMC was 160 (mg/kg-1), and the comparative efficacy dose was intravenous HYA20 (mg/kg-1) and oral HC@HMC (high dose) 160 (mg/kg-1), which means that the dosage of HYA was small when the drug was effective, so there was a significant difference between the two. If the concentration of drugs in the brain is measured, when the intravenous dose is 20 (mg/kg-1), the concentration in the brain is lower, and if the oral dose is still 160 (mg/kg-1), the concentration in the brain is much different.
Questions about B/P ratio,Please see the attachment.

Reviewer 4 Report
Thanks that authors tried to address some previous comments. However, the major comment that there is no evidence/data to demonstrate the nanoparticles is not addressed. Again, authors mixed many excipients together, but they do not know if they got nanoparticles or not. The whole study is about nanoparticles. Without data for nanoparticles, this study cannot be published.
In addition, just based on a slower clearance or clearance change to determine nanoparticle presence in blood is not correct. Excipients could change PK profiles, not just nanoparticles.
Author Response
Thanks for your kind suggestions.
To demonstrate the formation of nanoemulsion, the particle size and potential of HC@HMC were measured by the Malvern laser particle size potentiometer. The average Zeta potential of HC@HMC was about 0 mV, but the particle size result was not obtained. According to the literature, the particle size value of water-in-oil nanoemulsion is "abnormal" : For example, Hu et al. [1] diluted evodiamine water-in-oil nanoemulsion with ethyl oleate, and the particle size measured by quartz sample cell was 554 nm, and the particle size measured by polystyrene sample cell was 2789 nm. The difference between the two is obvious, which shows that the sample cell material has a great influence on the sample particle size measurement. We could not measure the particle size of HC@HMC, possibly due to the fact that our drugs, excipients, preparation processes and sample cell materials are different from those reported in other literature. However, it can be seen from the appearance diagram that HC@HMC is a clear liquid with uniform dispersion, which can preliminarily indicate the successful preparation of HC@HMC. The obtained nanoemulsion HC@HMC also possessed the basic properties of nanoemulsion: stability and uniformity. In addition, we believe that this study is valuable because the results suggest that HC@HMC can improve the bioavailability of HYA and thus the therapeutic efficacy of HYA.
Ref.
[1] Hu J, Sun L, Zhao D, et al. Supermolecular evodiamine loaded water-in-oil nanoemulsions: enhanced physicochemical and biological characteristics. Eur J Pharm Biopharm. 2014;88(2):556-564. doi:10.1016/j.ejpb.2014.06.007